# Stratosphere-troposphere exchange in the vicinity of a tropopause fold

Christiane Hofmann<sup>1</sup>, Astrid Kerkweg<sup>1</sup>, Peter Hoor<sup>1</sup>, and Patrick Jöckel<sup>2</sup>

<sup>1</sup>Institute for Atmospheric Physics, Johannes Gutenberg University, Mainz, Germany <sup>2</sup>Deutsches Zentrum für Luft-und Raumfahrt (DLR), Institut für Physik der Atmosphäre, Oberpfaffenhofen, Germany *Correspondence to:* C. Hofmann (hofmach@uni-mainz.de)

**Abstract.** Transport of air masses from the stratosphere to the troposphere along tropopause folds can lead to peaked ozone concentrations at ground level and hereby influence the long-term trend of tropospheric ozone. To improve the understanding of responsible processes and preferred regions of exchange, transient and reversible exchange processes in the vicinity of a tropopause fold are analysed on the basis of a case study. The global and regional atmospheric chemistry model system

- MECO(n), which couples the limited-area atmospheric chemistry and climate model COSMO-CLM/MESSy to the global model ECHAM5/MESSy for Atmospheric Chemistry (EMAC) is applied. Using similar process parametrisations in both model instances, the system allows for very consistent, simultaneous simulations at different spatial resolutions. Simulated ozone enhancements at ground level, caused by descending stratospheric air masses, are evaluated with observational data. Because of the coarse resolution of the global model, the observed ozone enhancements are not captured by the global model
- instance. However, the results of the finer resolved, regional model instance coincide well with the measurements. Based on the combination of Eulerian and Lagrangian analysis methods it is shown that stratosphere-troposphere-exchange (STE) in the vicinity of the tropopause fold occurs in regions of turbulence and diabatic processes. Within the framework of a Lagrangian study the efficiency of mixing along a tropopause fold is quantified, showing that almost all (97%) of the air masses originating in the tropopause fold are transported into the troposphere during the following two days.

# 15 1 Introduction

Tropospheric ozone is well known as an important oxidant and a precursor for highly reactive radicals. It is one of the most important greenhouse gases and toxic to all living creatures in case of high concentrations occurring at the Earth' surface (e.g., Bell et al., 2004; Solomon et al., 2010). Although a large fraction of tropospheric ozone is produced photochemically, significant amounts of ozone have stratospheric origin (e.g., Stevenson et al., 2006; Lefohn et al., 2011). Depending on the

20 resolution of the underlying data and the tropopause definition (e.g., Gray, 2003; Rind et al., 2007), estimates of the global, annual ozone contribution of stratosphere-troposphere-exchange (STE) vary between 340 Tg and 1440 Tg, corresponding to 5% to 20% of total production of tropospheric ozone (Wild, 2007). Since the maximum contribution of stratospheric ozone in the extratropics occurs in winter and spring, when tropospheric ozone has a longer life time, the stratospheric contribution is estimated to be even higher (Stohl et al., 2003). Roelofs and Lelieveld (1997) for instance estimate the stratospheric contribution

to tropospheric ozone at 40% at the maximum. Because of enhanced wave breaking of synoptic waves due to global warming, this contribution of STE to tropospheric ozone is estimated to increase during the next decades (Stevenson et al., 2006; Hegglin and Shepherd, 2009).

Various studies on the global freuqency of STE show that locations, where transport from the stratosphere into the troposphere (STT) occurs, are zonally and seasonally highly variable (Wernli and Bourqui, 2002; Sprenger and Wernli, 2003; Skerlak et al., 2014). For the extratropics, the cross tropopause mass flux is largest during winter, spring and autumn along the northern hemispheric cyclone tracks (Sprenger and Wernli, 2003). Reutter et al. (2015) recently quantified the importance of extratropical cyclones, showing that roughly 50-60% of all STE events over the North-Atlantic occur in the vicinity of them. Deep STT events, where stratospheric air parcels reach tropospheric levels below 700 hPa, are concentrated over the
continents, the preferred region for baroclinic wave breaking (Skerlak et al., 2014). Sprenger and Wernli (2003) estimate that

50-70% of the cross tropopause mass flux in the extratropics can be attributed to tropopause folds.

The propagation of the air parcels during deep STT events can be divided into three phases (Bourqui and Trepanier, 2010): (1) air parcels cross the tropopause, (2) descend quasi-adiabatically (loosing only 1 K/d) along the tilted isentropes before they (3) disperse in the lower troposphere depending on the current synoptic situation. Tropopause folds developing in the

15 baroclinic region below the jet stream (Keyser and Shapiro, 1986), have already been identified as preferred STT region by Shapiro (1980). However, the quantification of the importance of STE in the vicinity of tropopause folds and the responsible processes are still highly uncertain and a current topic of research (Lefohn et al., 2011; Lin et al., 2012; Skerlak et al., 2014; Lin et al., 2015; Tyrlis et al., 2014; Skerlak et al., 2015, and references therein).

To seperate stratospheric and tropospheric regions in the vicinity of tropopause folds, commonly the well-established dynam-

- ical tropopause is applied (e.g., Holton et al., 1995; Wernli and Bourqui, 2002). Depending on season, potential temperature and geographical latitude, the value of potential vorticity (PV, Ertel, 1942) used for the definition of the tropopause in the extratropics varies between 1.5 and 5 pvu (1 pvu =  $10^{-6} \text{ m}^2 \text{s}^{-1} \text{K} \text{kg}^{-1}$ ) (Kunz et al., 2011). Due to its stable stratification, the stratosphere is characterised by high PV values, whereas PV in the troposphere is around 1 pvu on average. Thus, crossing the dynamical tropopause from the stratosphere to the troposphere requires non-conservative processes, leading to the decrease of
- PV. Shapiro (1980) concluded that turbulence is the first order process, leading to STT along tropopause folds. Furthermore, diabatic processes, like cloud latent heating (Bourqui, 2006), cloud top radiative cooling (Wirth, 1995; Forster and Wirth, 2000) or convective activities (Cooper et al., 2005) can also be responsible processes for STT.

Various methods have been developed in recent years to analyse upward and downward mass fluxes across the tropopause separately in results of simulations conducted with global or regional models. While purely Eulerian approaches (e.g., Wei,

1987) are fraught with inaccuracies, for instance due to interpolation errors (Gettelman and Sobel, 2000), Lagrangian techniques are affected by errors due to numerical diffusion. Thus, artificial fluxes across the tropopause, which are not necessarily caused by physical processes, might also be considered. Therefore, Wernli and Bourqui (2002) introduced the residence time criterion to eliminate STE events occuring due to non-physical processes. However, in that case the calculation of the cross tropopause mass fluxes strongly depends on the choice of the residence time (Bourqui, 2006).

In addition to these general uncertainties of Eulerian and Lagrangian analysis methods, the horizontal resolution of atmospheric chemistry models is a limiting factor for the detailed analysis of STT events in the vicinity of tropopause folds (Gray, 2003). Based on a model comparison study, Cristofanelli et al. (2003) concluded, that it is impossible to capture deep STT events in models with horizontal resolutions coarser than 1.85°.

- In recent years, new approaches have been developed to better represent the ozone contribution of STT and particulary the ozone distribution at ground level in Eulerian models. Zhang et al. (2011) achieved an improvement in simulated surface ozone, when they used a regional, higher resolution model in addition to the global GEOS-Chem global 3-D chemical transport model (http://www.geos-chem.org/). However, the model significantly underestimates events of exceptionally high ozone occuring due to stratospheric intrusions. The application of a coupled system of GEOS-Chem and regional instances
- of the Comprehensive Air quality Model with extensions (CAMx, http://www.camx.com/) qualitatively reproduces observed surface ozone enhancements from stratospheric intrusions, but underestimates the total amount of ozone from STE (Lefohn et al., 2013). Lin et al. (2012) showed that it is possible to simulate temporary ozone enhancements, caused at the surface by descending stratospheric air masses, using the chemistry climate model GFDL AM3. However, the correct time and location of the enhancements are not reproduced correctly. A better temporal and spatial agreement is achieved with the Lagrangian anal-
- ysis technique: Lefohn et al. (2011) were able to associate measured ozone enhancement at observation stations to descending stratospheric air masses.

For the present study, the Eulerian as well as the Lagrangian technique are used to analyse ozone enhancements at ground levels. The aim of this analysis is to quantify the stratospheric ozone contribution and to identify responsible processes leading to STT in the vicinity of a tropopause fold. The applied model system MECO(n) (MESSy-fied ECHAM and COSMO models

nested n-times) and its configuration are described in Sect. 2. Section 3 gives an overview of the synoptic situation of the case study. The contribution of STT to surface ozone concentrations and responsible processes for STT are analysed in Sect. 4. This section starts with an evaluation of the different resolved model instances (Sect. 4.1), before the exchange processes are analysed in detail in Sect. 4.2. The efficiency of mixing in the vicinity of a tropopause fold is investigated in Sect. 5, before all results are summarised in Sect. 6.

# 25 2 Model description and set-up

The model system MECO(n) contains two different base models: the global model ECHAM5/MESSy for Atmospheric Chemistry (EMAC, Jöckel et al., 2010) and the regional atmospheric chemistry and climate model COSMO-CLM/MESSy (hereafter referred to as COSMO, Rockel et al., 2008; Kerkweg and Jöckel, 2012a). Each base model is denoted as one instance of the model system MECO(n). The coupling between the global EMAC instance and an arbitrary number of COSMO instances is

30 done on-line using the Multi-Model-Driver software (MMD, Kerkweg and Jöckel, 2012b), which is part of the Modular Earth Submodel System (MESSy). The MESSy software (Jöckel et al., 2010; Jöckel et al., 2015) currently provides about 60 submodels, which can be subdivided in submodels for infrastructure, diagnostics, atmospheric chemistry and model physics. Since MESSy is implemented in EMAC (Jöckel et al., 2005) as well as in COSMO (Kerkweg and Jöckel, 2012a), the same chemistry

mechanisms and process parametrisations can be applied for both base models, leading to a high consistency between both model instances.

# 2.1 Model configuration and meteorological initialisation

In this study a MECO(1) set-up is used, which means one COSMO instance is nested into the global EMAC (Fig. 1). The finer

- 5 resolved COSMO instance has a grid point distance of  $0.125^{\circ}$  ( $\approx 14$  km), 40 levels in the vertical and a time step of 40 seconds. EMAC is applied at T106L31 resolution ( $\approx 90$  km) with a time step of six minutes. The vertical grids of both instances reach up to 10 hPa. The EMAC instance is initialised by ECMWF analysis data interpolated to the EMAC grid. Since this study is focused on a specific meteorological situation, a weak nudging of four prognostic variables (temperature, divergence, vorticity and the logarithm of surface pressure) towards ECMWF analysis data is applied (described by Jöckel et al., 2006). During the
- 10 initialisation phase the initial fields of EMAC are interpolated to the COSMO grid using INT2COSMO. Boundary data for the COSMO instance are provided at every EMAC time step.

The case study investigated in this study comprises the development of a low pressure system in March 2010 (described in Sect. 3). Two simulations with different start dates and tracer initialisations have been performed. To analyse the impact of stratospheric air masses on surface ozone concentrations (Sect. 4), a simulation including chemistry (described in Sect. 2.2)

has been started at 24 March 00 UTC. In contrast, the investigation of the efficiency of mixing along the tropopause fold (Sect.
requires a longer simulation time, while a full chemistry mechanism is not necessary. Therefore, this simulation starts at 23 March 00 UTC and only passive tracers (described in Sect. 2.2) are included. Both simulations end at 28 March 00 UTC.

# 2.2 Atmospheric chemistry set-up and initialisation

The chemical tracers are initialised using the results from ESCiMo simulation RC1SD-base-10a (described in detail by Jöckel
 et al., 2015). In this simulation, EMAC has been applied with a resolution of T42L90MA. To avoid discrepancies that might occur during the first hours of a chemistry simulation which is initialised with coarser resolved data, an intermediate EMAC simulation in T106L31 has been started four days before the beginning of the MECO(1) simulation.

For the chemistry calculation, applied submodels and their settings have been adopted from the RC1SD-base-10a reference simulation. Since halocarbon chemistry in the troposphere can be neglected on timescales of days, it has been excluded from the chemistry mechanism.

In addition to the chemical tracers, a passive ozone tracer is initialised at the beginning of the MECO(n) simulation. In contrast to the chemically active ozone, passive ozone is only transported (advection, convective and diffusive transport) and not affected by any chemical reactions or depletion processes. Using the submodel PTRACINI (see Appendix A), artificial, stratospheric tracers are released with an initial mixing ratio of 100 nmol/mol, fulfilling the criteria listed in Table 1. Two types

30 of these artificial tracers are distinguished: (1) tracers, which are released only at the beginning of the simulation and thus, fulfill the criteria only at simulation start and (2) tracers which are continuously re-initialised during the simulation in the area where the criterion is fulfilled. The latter tracers are additionally labeled with a "c".

# 3 Synoptic situation

In this study, the development of the low pressure system "Ingeborg" is investigated. It grows out of a strong low pressure complex east of Newfoundland on 23 March 2010 and reaches maximum strength at 25 March 06 UTC with a minimum sea level pressure of 969.5 hPa to the southwest of Ireland. At this time, its frontal system is already occluded and extends to the African continent. Ahead of the occluded system, warm subtropical air masses are transported northeastward from southerly

- regions, whereas cold and dry air masses descend from the north in its wake. The development of the strong baroclinic zone, which is shown in Fig. 2 for EMAC (top) and COSMO (bottom), is characteristic for the low pressure system "Ingeborg". On 26 March, the centre of the low pressure system breaks into two separate pressure minima: "Ingeborg I" dissipates over Southwest Britain on 27 March, whereas "Ingeborg II" initially stagnates over Scotland (Fig. 2). In the meantime, on 26 March
- a new low pressure system "Judy" develops along the baroclinic zone (Fig. 2, bottom). Associated vertical motion along the frontal system leads to strong precipitation events.

That meteorological situation has already been studied with the MECO(n) system (Hofmann et al., 2012). In this study, the model results have been compared with precipitation measurements and analysis data. It has been shown, that MECO(n) is able to capture the development of the low pressure systems "Ingeborg" and "Judy". Compared to the model set-up in

- Hofmann et al. (2012), in the present study a larger model domain as well as a coarser resolution for COSMO (0.125° instead of 0.0625°) is used. Despite of these changes, the meteorological situation over Europe is still captured well by both model instances. Fig. 2 shows the position of the low pressure system at 26 March 12 UTC compared to ECMWF analyses. Both model instances reproduce the low pressure systems approximately correctly: while the position of "Ingeborg I" and "Ingeborg II" is captured correctly in EMAC, the intensity and the development of "Judy" are represented better in COSMO. Large
- gradients in equivalent potential temperature, which are more concise in the COSMO instance, indicate the position of the frontal system, which reaches Germany around 26 March 12 UTC in agreement with the ECMWF analyses.

# 4 Contribution of STT to surface ozone concentrations and responsible processes for STT

#### 4.1 Comparison with surface ozone observations

Figure 3 shows the development of the mixing ratio of  $STRATO_{2pvu}$ , initialised as described in Table 1 as vertical cross sections along 45°N. While the tracer in COSMO is already transported to the Earth' surface one day after initialisation, it only reaches ground level in EMAC on the second day. Additionally, the mixing ratios at the surface are distinctly lower in EMAC compared to the results of COSMO. Differences are also obvious in the tropopause structure, which shows more small scale features in the COSMO instance.

Similar differences between both model instances occur regarding the tracer distribution at the lowest model level (Fig. 4).
 In EMAC as well as in COSMO, STRATO<sub>2pvu</sub> reaches the lowest model level around 20 hours after initialisation over the Atlantic. However, the mixing ratios in COSMO are distinctly larger and show finer structures and steeper gradients.

The comparison of the tracer transport with the synoptic situation (Fig. 2 and Fig. 4) clearly indicates that the stratospheric tracer reaches the surface behind the frontal system. As evident from Fig. 4, this frontal system acts as a barrier for the tracers, which are therefore only distributed behind the front, inside the cold sector of the cyclone. Because of the ongoing downward transport along the tropopause fold (Fig. 3), the largest mixing ratios occur exactly behind the front, leading to high gradients of the mixing ratios in that region.

In Fig. 5 time series of tracers at the position of the Jungfraujoch in Switzerland are shown. Due to its remoteness, ozone at this station is expected to be influenced rather by transport than by lower tropospheric sources. In both model instances, the stratospheric tracer  $STRATO_{2pvu}$  (top panels) is distributed all over the troposphere after 26 March 15 UTC. Apparently, only COSMO shows a temporary increase of the mixing ratio exceeding 18 nmol/mol within the entire troposphere. As already

- mentioned for the vertical cross sections (Fig. 3), COSMO shows again finer structures of the dynamical 2 pvu tropopause compared to EMAC (black contour), whereby the tropopause fold already dissipates when it reaches the Jungfraujoch. High values of PV (> 2 pvu) in the middle and lower troposphere are caused by diabatic processes; in this case the precipitation events are caused by raising processes along the frontal system (Sect. 3). In EMAC, these PV anomalies are lower (< 2 pvu, not shown), possibly because the frontal precipitation is less pronounced and located too far in the north in EMAC (Hofmann</p>
- et al., 2012). Considering the lower troposphere,  $STRATO_{2pvu}$  approaches the station level at 26 March 15 UTC in both model instances. While the mixing ratio at the surface in EMAC reaches maximum values of around 8 nmol/mol in the following five hours, the mixing ratio in COSMO increase up to 18 nmol/mol at the Earth' surface due to the temporary strong increase metioned above.
- In the bottom panel of Fig. 5 time series of ozone are shown. The comparison of the stratospheric tracer (top) and ozone (bottom) clearly indicates that the time period of enhanced stratospheric tracer and ozone coincide. Especially around 26 March 15 UTC, when the stratospheric tracer is transported into the lower troposphere, the structures of STRATO<sub>2pvu</sub> and ozone coincide very well, at least in COSMO. While this increase of the mixing ratio of the stratospheric tracer in the lower troposphere at 26 March 15 UTC is also visible in EMAC, the mixing ratio of ozone remains constant in the coarser instance. To evaluate the differences between both model instances, the mixing ratios of the artificial, stratospheric tracer and of ozone
- are compared with hourly surface measurements of ozone. The measurements are conducted by the EMEP network (European Monitoring and Evaluation Programme, www.emep.int) and are provided by the EBAS database (ebas.nilu.no). Beside the station in Switzerland (Jungfraujoch, JJ), measurements from Spain (Noya, NO), France (La Coulonche, LC) and Germany (Schauinsland, SC) are considered, as examples of different stations all over Europe. The location of the different stations are marked in the top left panel of Fig. 4. Tracer time series at these stations are shown in Fig. 6. Since the ozone measurements
- (green) are given in  $\mu$ g/m<sup>3</sup>, simulated tracer mixing ratios are converted accordingly. Beside the chemically active ozone (red), also a passive ozone tracer (described in Sect. 2.2, blue) is shown. For a better comparison, the artificial, stratospheric tracers STRATO<sub>2pvu</sub> (yellow) and STRATO<sub>4pvu</sub> (pink) are additionally scaled by a factor of 0.05 and 0.15, respectively. Different tracers are shown in different colours, whereas the darker colours always represent the results of EMAC.

At the time of the frontal passage (NO: 25 March 02 UTC, LC: 25 March 14 UTC, SC: 26 March 12 UTC, JJ: 26 March 35 15 UTC), measurements at all stations show an increase of observed ozone. By means of the stratospheric tracer, which also

show an enhancement at this time, it is possible to relate the observed increase to descending, stratospheric air masses. Since the increase is visible in both stratospheric tracers, it can be concluded that these air masses originate from stratospheric regions with PV > 4 at 24 March 00 UTC. Considering the ozone tracers, both model instances produce an increase associated with the frontal passage. In COSMO the amplitude and timing of this increase coincide better with the observation than in EMAC.

- For instance, in Noya at 25 March 02 UTC ozone increases by the amount of 10 μg/m<sup>3</sup> in EMAC, whereas the increase is larger than 20 μg/m<sup>3</sup> in COSMO, like in the observations. At the stations in La Coulonche and Schauinsland, the increase in EMAC is delayed: maximum ozone values in EMAC occur approximately three hours later compared to COSMO and the observations. Enhanced ozone in the troposphere, which is simulated before the frontal passage (e.g., Fig. 6 c, 25 March 14 UTC) is either produced by photochemical production or transported within the troposphere since 24 March 00 UTC (start of the simulation).
- Tracer masses of passive ozone are always larger than those of chemically active ozone because of missing chemical depletion processes. For the stations in Noya and at the Jungfraujoch, these differences are rather small, but there is a large discrepancy between simulated and observed tracer masses. Since both stations are located in rural regions, away from areas with high NO<sub>x</sub> emissions, they do not show any diurnal cycle. This is consistent with the measurements. Missing nocturnal depletion processes can therefore be excluded as reasons for these too high ozone values. In contrast, at the stations in La Coulonche and
- Schauinsland, the nocturnal depletion leads to a better agreement between the simulated and observed ozone masses, even if they are still slightly enhanced. These too high ozone values are a consequence of the initialisation: in both model instances ozone is initialised using the ESCiMo simulation RC1SD-base-10a, where ozone is known to be overestimated (Jöckel et al., 2015). However, the relative enhancements can be attributed to the STT events.

To present a more objective analysis, the correlation between the observed and simulated ozone has been calculated (Table

2). Since this study focuses on surface ozone enhancements associated with the frontal passage, a 12 hour time window around the frontal passage at the relevant station (NO: 25 March 02 UTC, LC: 25 March 14 UTC, SC: 26 March 12 UTC, JJ: 26 March 15 UTC) is considered. Hereby, the qualitative results from the above can be confirmed: at all stations, the results from COSMO correlate better with the observations compared to the EMAC results.

Since this evaluation has shown that only the COSMO instance is capable to capture the descent of stratospheric air masses into the troposphere in accordance to measurements, for the analyses hereafter, only the results of COSMO are considered.

#### 4.2 Detailed analysis of the exchange processes

Due to the good agreement between the results of the COSMO instance and the observations found in Sect. 4.1, the results of COSMO are now used to further analyse the STT events along the tropopause fold in detail. Applying the trajectory model LAGRANTO (Lagrangian Analysis Tool, Wernli and Davies, 1997) to the COSMO model output, backward trajectories are

calculated to obtain information about (1) the origin of the air masses responsible for the observed surface ozone enhancement at the time of the frontal passage and their development previous to STT, (2) responsible processes for STT, and (3) the location of STT and the propagation in the troposphere.

To capture the STT events leading to the high surface ozone values, the backward trajectories start at 26 March 12 UTC, the time of the frontal passage in Central Europe. They are initialised, whereever the mixing ratio of  $STRATO_{2pvu}$  is larger

than 10 nmol/mol in the lower troposphere (below 3000 m, see Fig. 4, greenish area in bottom right panel) and calculated backward for 59 hours (until time of first data output of simulation). To identify trajectories, which experienced a transition from the stratosphere to the troposphere, only those trajectories with PV < 2 pvu at 24 March 01 UTC are selected. Futher on, the position of STT (PV < 2 pvu first-time) along each trajectory is determined, where trajectories returning temporarily into the stratosphere are neglected.

# 4.2.1 The origin of the air masses and their development previous to STT

As an example, Fig. 7 shows the behaviour of trajectories with STT at 24 March 12 UTC (left) and 25 March 00 UTC (right). All trajectories start bundled to the east of Newfoundland and run southeastward over the North Atlantic. During that time the air parcels have stratospheric PV values (> 2 pvu) and are located at pressure levels around 500 hPa (Fig. 7 a,b). The

- pathway in the stratosphere before the STT event coincides with the horizontal position of the jetstream (black contour in Fig. 10 7 c,d), which reaches maximum wind velocities of 70 m/s at 300 hPa (Fig. 7 e,f). The air parcels therefore originated in a stratospheric region below the jet stream and descend deeper into the associated tropopause fold, initially isentropic along the tilted isentropes.
- Notably, inside the tropopause fold the PV along the trajectories increases, although the air parcels descend (Fig. 7 g,h). As indicated in Fig. 7 (c-f) by enhanced values of the turbulence index (TI, Ellrod and Knapp, 1992), turbulence occurs 15 inside almost the entire fold (cyan contour). Thus, turbulence might play a major role for the initial PV modification inside the tropopause fold. As diffussive (i.e., non-adiabatic) process it can potentially modify the vertical gradient of the diabatic heating rate which constitutes one part of the PV-tendency equation. Approximatly at the time the air parcels experience maximum PV and TI (g: 24 March 06 UTC, h: 24 March 12 UTC), the potential temperature as well as the PV begins to decrease, leading to the transition into the troposphere a few hours later. 20
  - 4.2.2 **Responsible processes for STT**

# The strongest decrease of potential vorticity can be observed in the two hours before and after the STT time (red contour in Fig. 7 g, h). As a result of the definition of PV, non-conservative diabatic as well as turbulent processes can be responsible processes, leading to PV change. Figures 7 (g) and (h) also show the continous decrease of potential temperature along the trajectories, with largest gradients (-4.3 K / 12 h for trajectories with STT at 24 March 12 UTC and -3.4 K / 12 h for those with STT at 25 24 March 12 UTC, resp.) during the hours before STT (purple contour). This strong diabatic cooling inside the tropopause fold might be caused by radiative processes, occurring in thin layers above the clouds and amount up to 10 K/d (Slingo et al., 1982). Because of these cooling processes, the air parcels descend and reach isentropes, which are already located entirely in the troposphere. During (left) and after (right) the transition into the troposphere, most of the air parcels enter this cloudy

region, which has been the reason for the decrease of PV previous to the STT event (orange contour). Beside these radiative 30 and therefore diabatic processes, turbulence can also lead to a change in PV. While turbulent processes have been identified to be responsible for the PV increase along the trajectories when air parcels initially enter the tropopause fold, they can also lead to PV reduction. Since the TI is still enhanced at the time of STT, turbulence might also contribute to the PV decrease. For the

case study presented here, it can therefore be concluded, that turbulent as well as diabatic processes are causing the STT events along the tropopause fold.

# 4.2.3 The location of STT and the propagation in the troposphere

- Because of the propagation of the tropopause fold, the positions of STT (crosses in Fig. 7 a-d) at 25 March 00 UTC are
  shifted southeasterly compared to twelve hours before. STT occurs below the jet stream, in a region with a low tropopause and enhanced turbulence (Fig. 7 c,d). All STT points (crosses in Fig. 7 a-d) are located at the northwesterly side of the tropopause fold. Thus, the air parcels leave the fold in the direction of the movement of the fold. The vertical location of STT is depicted in Fig. 7 (e) and (f) by crosses, exemplarily for 30°W (left) and 21°W (right), respectively. STT points are distributed along the northwesterly side of the tropopause fold, many of them reach the tropopause at levels below 700 hPa and are therefore directly
- located in the lower troposphere after STT. The colour of the crosses indicates the PV value, the corresponding trajectory started with at 24 March 01 UTC. It shows that even trajectories entering the troposphere at the tip of the fold have stratospheric origin  $(PV_{start} > 3 pvu)$ .

After entering the troposphere, transport proceeds almost quasi-isentropically. The baroclinicity in the lower troposphere and the associated tilted isentropes lead to further descent of the air parcels. The direction of movement of the air parcels in the

15 troposphere is determined by the synoptic situation, especially the wind field at time of STT. In this case study, the air parcels are therefore transported further southeast after STT, before they spread behind the frontal system.

# 5 On the efficiency of mixing along a tropopause fold

In this section the history and fate of the air masses inside the well developed tropopause fold are studied. The focus is on air masses below 6000 m height, which exhibit high ozone and PV. From a dynamical point of view, these air masses can potentially return to the stratosphere and would therefore not contribute to tropospheric ozone. The aim of the following analysis is to obtain information about the composition of the air masses inside the tropopause fold, their origin and destination regions. Therefore, a 5-day MECO(1) simulation with passive tracers has been initialised already at 23 March 00 UTC (described in detail in Sect. 2.2), allowing for the investigation of the STE events already on 24 March. First, 48h-forward-trajectories are initialised at grid points which are located inside the fold deep in the troposphere, but are stratospheric from a dynamical point

- of view. These air masses are selected according to the following criteria between 24 March 01 UTC and 25 March 23 UTC: PV > 2 pvu,  $STRATO_{c,2pvu} > 80 nmol/mol$ , height 

region somewhere inside the model domain, are located inside the pronounced tropopause fold or in finer filaments 24 hours later (Fig. 8) and are again 48 hours later located in their destination region.

To obtain the amount of air parcels transported out of the tropopause fold into the troposphere, the PV at the destination region is determined. If it is smaller than 2 pvu, the destination region is denoted as tropospheric (DT), otherwise as strato-

- spheric (DS). Similarly, the origin region three days earlier is determined (OT resp. OS). Absolute and relative numbers for this analysis are summarised in Table 3. The first column refers to trajectories located inside the tropopause fold at 25 March, 12 UTC along 40°N, exemplarily shown in Fig. 8. For this date and latitude, in total 769 trajectories are calculated, at which 756 of these (i.e., 98.3%) end in the troposphere two days later. 26.5% of these 98.3% in turn have their origin region in the stratosphere three days earlier, at which 73.5% are started in the troposphere. It is also possible to assess the composition of the
- air masses inside the tropopause fold: 73.6% of the air masses, located inside the tropopause fold at 25 March 12 UTC along 40°N have tropospheric origin, whereas 26.4% originated in stratospheric regions.

For the entire model domain, 7765.6 starting points per hour are identified on average, leading to a trajectory set of about 380000 trajectories for the entire study. The results are presented in the middle column of Table 3. Since different definitions of the dynamical tropopause exist in the literature, the sensitivity of this study is tested, using an extended tropopause criterion

- (right column of Table 3): the location of an air parcel is denoted as stratospheric, if the PV is larger than 4 pvu and as tropospheric if it is less than 2 pvu. That way, a new transition region originates, which can be seen as a tropopause band (TB), comprising all air masses exhibiting PV values between 2 and 4 pvu. The use of a tropopause band also accounts for sporadic exchange events, which do not lead to large changes of PV or insignificant exchange events, since PV changes are required to exceed at least 2 pvu.
- In agreement with the example in the first column, almost all air parcels (97.7%) located inside the tropopause fold at a certain time, are transported into the troposphere during the following two days. This is valid independently of the the origin of the air parcels: considering those with stratospheric origin, 97.2% are transported into the troposphere for instance. Applying the 2 pvu-criterion, around one third of the air parcels (32.1%) located inside the tropopause fold at a certain time have stratospheric origin, whereas two third (67.9%) originated in the troposphere. The amount of air parcels with stratospheric
- origin is smaller (4.5%), if the tropopause band between 2 and 4 pvu is considered. Thus, 27.7% of the air parcels are located in the tropopause band already 24 hours before their residence in the tropopause fold.

Beside the origin and destination region, also the frequency of STE events along a tropopause fold is analysed (Table 3, lower part). Regarding all trajectories of air parcels located inside the tropopause fold at a certain time, along most of them (67.4%) so-called transient events (origin and destination are the same) occur. This amount is considerably enhanced (95.6%),

30 if the tropopause band is considered. Almost all (98.6% for the 2 pvu tropopause criterion and 69.5% for the tropopause band criterion) of these air parcels, start and end in the troposphere. Regarding the non-transient events, which means the air parcels indeed leave their origin region, most (95.6% resp. 76.5%) of these exchange events in the vicinity of the tropopause fold are STT events. These STT events are, despite of their low percentage compared to the non-transient events, responsible for the observed ozone enhancements analysed in Sect. 4.