# Peer review of "Stratosphere-troposphere exchange in the vicinity of a tropopause fold"

_Atmospheric Chemistry and Physics, 2015_

## Referee Comment (RC1) · A. Stohl (Referee) · 25 Jan 2016

This modelling study of an STE event is, in general, relatively well written and quite clear. While I do think that several sections could be improved by more accurate and more detailed analysis (see points below), overall the analysis methods are solid and the combination of Eulerian and Lagrangian analyses of some interest. So there is nothing really wrong with this paper. However, my main point of criticism is that it simply does not add much to our general understanding of stratospheric intrusions. Similar studies of similar events have been published in great number and I can't find any novel results of this study in particular. Neither are the methods applied particu-larly novel (certainly good tools have been used) nor are the results showing something

that has not been shown before. Partly this is also related to the fact that this is a pure modeling study that does not involve any in-depth analysis of measurement data (the passing mentioning and use of ozone time series of surface stations does not really provide much extra insight). Often it is the combined use of models and measurement data that can reveal interesting aspects, whereas it is difficult to show something definitive if only a model is used. Overall, I think it is an editorial decision whether such a type of paper is of sufficient novelty to justify publication in ACP, or not. In my overall ranking, I suggested that the paper should be reconsidered after major revisions, as a compromise. But I think the decision is really an editorial one between acceptance with relatively minor revisions, or rejection because of lack of novelty. More specific comments about the paper are given below.

The episode chosen does not appear to be one with major surface ozone enhancements. There exist much stronger episodes with increases due to stratospheric intrusions. Why was this particular one chosen for a case study? Is it just because simulations already existed (and have been published already)?

Section 4.2.2 is not very clear and rather vague. It is touched upon which processes are responsible for STT, but they are not isolated and quantitatively discussed. In a pure model study, it should be possible to say which exact model process(es) is (are) causing the STT. There seems to be also some lack of a clear understanding about processes that can lead to STE. For example when it is written on page 11, l17, that your results indicate "that particularly non-conservative processes enable air parcels to cross the tropopause". Given the definition of your tropopause in PV units, only diabatic and frictional processes can actually lead to such an exchange, and this is by definition so, not because your results indicate that. If they would indicate otherwise, something would be wrong.

Section 5: For my taste, there are too many relative values here, making the whole section somewhat difficult to read, especially when relative values of relative values are given. Furthermore, it would be interesting to know what the actual mass exchange rates (or air and of ozone) are, as they could be compared to other cases documented in the literature. The relative frequencies do not suffice in that regard. Surprisingly, then, the last sentence of your paper says that "coarse resolution models might underestimate STE mass and ozone fluxes" – so why was this not shown in the first place, especially since you have done your study with one coarse- and one high-resolution model. With accurate analysis, this statement could be quantitative rather than speculative (at least for the event studied).

Further specific points:

To my (admittedly, personal) taste, the use of names for individual modules or subroutines is inflationary. This creates chains of acronyms that may sound funny but are actually difficult to remember and unnecessarily extend the text. This concerns many things in this paper but one example is the "submodel" PTRACINI that seems to be nothing more than a subroutine in a computer code that does some tracer initializations. Does it really need its own name, suggesting that this is something of greater significance?

Somewhat related to the above point, is it necessary to explain MECO(n) in a general way if anyway only one nesting level is used, as explained on page 4?

Pg 2, lines 30-31: There are a few strange statements about problems with Lagrangian and Eulerian approaches. For instance, numerical diffusion is highlighted as a problem for Lagrangian models, whereas it is exactly the (near) absence of it that makes Lagrangian models valuable tools. For Eulerian models, interpolation is highlighted as a problem, which also is not true.

Pg 5, l7: Where is it actually shown that this zone is baroclinic? General synoptic understanding suggests that but it is not really shown, is it?

Fig. 2: It appears that EMAC SLP agrees better with ECMWF analysis than COSMO SLP. Any good explanation for that?

Pg 6, line 2: Be concrete and accurate: Which front? You mean the cold front but this should be said explicitly.

Pg 6, l6: Jungfraujoch is described as a remote site but it is located in the heart of Europe and also sees regular pollution episodes, especially in summer, e.g., from the Po Valley.

Pg 6, l13: "raising processes": What shall that be? Do you mean convection? Or slantwise frontal ascent? Again, language not very precise.

Figure 7g-h: I understand that the normalized scale may make these panels more compact, but it also hides a lot of information. For example, what is the actual change in potential temperature? This is referred to e.g. on page 9, l13, but cannot really be judged well from these figures. It should be possible to plot this also with absolute scales without inflating the panels too much.

Language issues:

Pg 1, l12: vicintity -> vicinity

Pg 2, l14: use of word "current" not correct here

Pg 3, l7: twice global within a few words

Pg 3, l9: occuring -> occurring

Pg3, l13: correct superfluous, as there is also the word "correctly" in same sentence.

Pg 6, l17: increaseS

Pg 7, l1: showS

Pg 7, l4: observationS

Pg 7, l9: "produced by photochemical production" - > produced photochemically

Pg 8, l12: "initially isentropic along the tilted isentropes" -> initially following the tilted

isentropes".

Pg 8, l18: ApproximatEly

Pg 10, l21: the the (remove on "the")

Pg 10, l24: two thirdS

Pg 10, l24: amount -> fraction

Pg 11, l6-7: Sentence starting with "By now, . . ." is unclear. Not sure what you want to say here.
* * *

---

## Author Comment (AC1) · 5 Feb 2016

We thank Andreas Stohl for his report which contains many helpful comments and suggestions for improvements. We will revise our article with respect to his points and answer open questions soon. However, we would like to comment on his main point of criticism in advance:

- *However, my main point of criticism is that it simply does not add much to our general understanding of stratospheric intrusions. Similar studies of similar events have been published in great number and I* **can't find any novel results**

*of this study in particular.*

Reply:

Indeed, there are many relevant studies of stratospheric intrusions available. However, all of them show that models are still not able to reproduce observed ozone enhancements at the surface caused by stratospheric intrusions correctly in time, location and amount, which shows that there is further need for improvements.

Furthermore, as far as we know, this is the first time the efficiency of mixing along a tropopause fold has been analysed. To our knowledge there is no other study, which addresses the question of irreversibility and transience of exchange associated with a fold. We quantified the fraction of air masses, which are indeed transported into the troposphere as well as the composition of the air masses inside the tropopause fold.

- ***Neither are the methods applied particularly novel*** *(certainly good tools have been used) nor are the results showing something that has not been shown before.*

  Reply:

  We present the results of a newly developed model system, which allows for consistent simulations on different scales. Therefore, this is the first time chemical tracers on global and regional scale can be directly compared, i.e. using the same model parametrisations, chemistry calculations, initialisation criteria, etc. for the global and the regional model.

  Since in earlier studies the impact of stratospheric intrusions on surface ozone concentrations has been analysed either from Eulerian or from Lagragian point of view, the combination of both techniques might also be seen as a novel applied method.

  Furthermore, the initialisation of artificial tracers directly inside the tropopause

fold and subsequent analyses mentioned above have never been performed before.

- *Partly this is also related to the fact that this is a **pure modeling study that does not involve any in-depth analysis of measurement data** (the passing mentioning and use of ozone time series of surface stations does not really provide much extra insight). Often it is the combined use of models and measurement data that can reveal interesting aspects, whereas it is difficult to show something definitive if only a model is used.*
Reply:
The study was intended as a chemical process study to investigate the efficiency of mixing associated with the fold and its effect on surface ozone. For a full 3d-process study it is surprisingly difficult to find an appropriate data set including turbulence and wind measurements with high resolution tracer measurements of the required precision. This is of course the next step to go to investigate the physical processes and the relevant time scales which lead to the erosion of the fold. In the current manuscript we wanted to show the potential consequences of the different model resolutions for the correct simulation of the surface ozone fields.
With this study the potential of the model system MECO(n) has been demonstrated. It can now be applied to analyse e.g. aircarft measurement data in future studies.

---

## Referee Comment (RC2) · Anonymous Referee #2 · 26 Feb 2016

**Review for Stratosphere-troposphere exchange in the vicinity of a tropopause fold by Hofmann et al.**

**Synopsis:**

Hofmann et al. look at a tropopause fold event and study the mass exchange (STE) in its vicinity. The topic is of interest to the readership of ACP. Of course, this is not the first study considering STE and trying to understand the mechanisms leading to the crossing of the dynamical tropopause. Nevertheless, I think that the problem is by far not yet solved, and additional careful case studies can shed further light on the relevant processes. Given this, I would recommend the study for publication in ACP. However, I see some space for improvement, which I will list below.

**Major Concerns:**

- In Figs. 3,4 and 5 the EMAC and COSMO simulations are shown. However, I am not convinced that we can learn too much about the processes leading to STE by looking at the EMAC simulations. The horizontal resolution of the model is simply too coarse (as the authors themselves state in the conclusions) and therefore the relevant physical processes are not adequately represented. To me, only the high-resolution COSMO simulation is of interest and really allows to study the processes. In short, I would recommend not to compare the two models explicitly, and simply mention the EMAC simulation as an intermediate step. No EMAC figures need to be shown and the text related to the comparison can be shortened. Note that from section 4.2 on, the authors in fact discuss only the COSMO results!

- Table 3 and its discussion 'confused' me - less would be more. In fact, I see the benefit from distinguishing different source and target regions, but the contents in Table 3 is simply too much. For instance, I wonder what can be learned from only looking at the region 40 N (first column)? Or if there is a good reason to have the 2-4 PVU criterion (last column), why also looking at the strictly-2 PVU case (middle column)? Maybe, what I am was missing is a clear motivation for this whole detailed analysis. I think it would help a lot if from beginning some clear hypotheses could be stated, and Table 3 is then the data supporting (or rejecting) exactly these hypotheses. Now, while reading the text I had a little the impression that many of the entries/calculations in the table are simply there because they could easily be done. Of course, the authors might see it differently! Then, I only request a more concise discussion!

- At some places, the text reads too technical. I managed to keep track until section 2.2, but then came 'ESCIMo simulation RC1SD-base-10a', which I do not know. Furthermore, there is definitely no need to mention 'INT2COSMO', and the whole Appendix A 'PTRACINI' is rather technical (with namelists,...) which in my view is not relevant for the reader, or should be discussed in a non-technical way. In the same line: On P9,L25-26 the calculation of trajectories is described: First, 48-h trajectories forward in time; then - if still in the domain - 72-h backward trajectories from the end points; resulting in information of the initial trajectories 24-h before they are in the fold. Why not simply write that 24-h backward trajectories are also calculated from the initial position in the fold and then combined with the 48-h forward ones. This sound less complicated to me? Or do I not really understand the reason for the strategy presented in the paper?!

- Personally, I would restructure the study! First, in section 3 I do not see a benfit in discussing the EMAC simulation; then section 4.1 is rather long. It compares the simulations with observations. Its main goal is concisely summarized in the last sentence of the section: "Since this evaluation has shown that only the COSMO instance is capable to capture the descent of stratospheric air masses into the troposphere in accordance to measurements, for the analyses hereafter, only the results of COSMO are considered." Note that the aim of the study, as stated in the introduction, is "to quantify the stratospheric ozone contribution and to identify responsible processes leading to STT". In fact, I think we do not really come closer to these aims with the rather detailed comparison with the observations. I would suggest to shorten this section 4.1 and then to combine it even with section 3. The new section would then define the synoptic and observational basis for the following process-related analysis. Then, I would suggest to split the new section 4 (with a potential title "Detailed analysis of the exchange process") more strictly into three parts: the first looks at the origin (near the jet) of the STE trajectories, the second handles with the STE parcels within the tropopause fold (which is rather interesting), and the third addresses explicitly the crossing of the dynamical tropopause. If the authors agree, this structure would more clearly follow the air parcel on their way from the stratosphere to the troposphere. But I leave it to the authors to decide whether they want to accept it or not.

- A remarkable result is the strong concentration of the stratospheric tracers behind the fronts (see Fig. 4). In the text the authors discuss the importance of fronts as 'barriers' for the tracers (P6,L3-4). Hence, they see the key impact of the fronts on the surface impact of the stratospheric tracers. But the discussion of the front-associated circulation, e.g., vertical winds is missing completely. Because fronts are identified as an important feature, I would appreciate a more detailed discussion. For instance, does the surface tracer imprint coincide with the frontal vertical winds?